# Identification of plasma proteins associated with seizures in epilepsy: A consensus machine learning approach

Saman Hosseini Ashtiani[1,2]*, Sarah Akel[1,2], Rakesh Kumar Banote[1,2,3¤],
Fredrik Asztely[1,4], Johan Zelano[1,2,3]*

**1** Department of Clinical Neuroscience, Sahlgrenska Academy, University of Gothenburg, Sweden,
**2** Wallenberg Center of Molecular and Translational Medicine, Sahlgrenska Academy, University of
Gothenburg, Sweden, **3** Department of Neurology, Sahlgrenska University Hospital, Gothenburg, Sweden,
**4** Department of Neurology, Angered Hospital, Gothenburg, Sweden

¤ Current address: AstraZeneca, Global Patient Safety BioPharma Organization, Chief Medical Office
R&D, 43153 Gothenburg, Sweden
* saman.ashtiani@gu.se (SHA); johan.zelano@neuro.gu.se (JZ)

doi.org/10.1371/journal.pone.0327317

Instiute, UNITED STATES OF AMERICA

**Peer Review History:** PLOS recognizes the
benefits of transparency in the peer review
process; therefore, we enable the publication
of all of the content of peer review and
author responses alongside final, published
articles. The editorial history of this article is
available here: https://doi.org/10.1371/journal.
pone.0327317

## Abstract

Blood-based biomarkers in epilepsy could constitute important research tools
advancing neurobiological understanding and valuable clinical tools for better diag-
nosis and follow-up. An interesting question is whether biomarker patterns could con-
tribute additional understanding compared to individual marker values. We analyzed
OLINK proteomics data from a large epilepsy cohort in which we have previously
found four differentially expressed proteins (CDH15, PAEP, LTBP3, PHOSPHO1).
Using two machine-learning techniques, we identified ten consensus candidate
protein biomarkers (CDH15, PAEP, LTBP3, PHOSPHO1, NEFL, SFRP1, TDGF1,
DUSP3, WWP2 and DSG3) that contributed to the classification of patients as being
seizure-free or not. Six out of the ten consensus proteins were identified as differen-
tially expressed in our previous study (although NEFL and TDGF1 not significantly so
after multiple testing correction). The remaining four consensus proteins were newly
identified by machine learning and were chosen for detailed analysis. In compari-
son to the four significantly differentially expressed proteins (CDH15, PAEP, LTBP3,
PHOSPHO1), the newly identified consensus proteins (SFRP1, DSG3, DUSP3, and
WWP2) and in particular a combination of all eight proteins, outperformed individual
proteins in identifying individuals with recent seizures, highlighting the potential of
multi-protein profiles. These findings emphasize the need for integrative bioinfor-
matic approaches in epilepsy research and underscore the role of neuroinflammation
and immune pathways in epileptogenesis. Our results support the applicability of
plasma protein profiling for developing future blood-based tests for epilepsy seizure
prediction, diagnosis, and treatment. Further validations in independent cohorts are
required to establish these candidate biomarkers in clinical practice.

**Data availability statement:** The underlying data in this study are sensitive personal data according to the Swedish Data Protection Regulation. The ethical approval of the Swedish Ethical Review Authority for this study (2020-853) states that the data and results can only be published on a group level, i.e., without any patient-level details. Both original and translated ethical approvals are uploaded. The contact details for the Swedish Ethical Review Authority that has imposed such restrictions along with a link to clear explanations identifying any patient-level health-related data as sensitive: registrator@etikprovning.se Phone: +46104750800 https://www.imy.se/en/organisations/data-protection/data-protection-within-different-areas/processing-of-personal-data--for-researchers/ Post address: Etikprövningsmyndigheten Box 2110 750 02 Uppsala

**Funding:** This study was funded by grants from Jeansson foundation, Swedish Society for Medical Research (SS18-0040), Region Västra Götaland (VGFOUREG-968476), Swedish state under the agreement between the Swedish government and the county councils and the ALF-agreement (715781).

**Competing interests:** J.Z. received speaker honoraria from UCB and Eisai for non-branded education events; and as employee of Sahlgrenska University Hospital is or has been an investigator/sub investigator in clinical trials sponsored by GW Pharma, SK life science, UCB, Angilini Pharma, and Bial (no personal compensation).

## Introduction

The field of epilepsy suffers from a lack of biomarkers for disease monitoring and neurobiological study. With increased biochemical possibilities to study brain health by blood samples, there is an increasing interest in identifying new blood biomarkers that either reflect seizure burden or can be used to understand pathological processes in epilepsy [1]. Several studies point to inflammatory markers being detectable in body fluids like plasma in active epilepsy. Clinically, symptoms of seizures can be subtle or missed entirely, and some patients are unable to communicate their seizure burden because of comorbidities.

Fluid biomarkers are not a new concept in epilepsy; seizures have long been known to leave metabolic traces in blood. Lactate is one of the more explored biomarkers of tonic-clonic seizures, and hormones like prolactin seem to increase after seizures [2–4]. Increased muscle enzymes can also be used to differentiate tonic-clonic seizures from syncope [3,5], but specificity and usefulness in other forms of epilepsy is unknown. Several reports also suggest that seizures, at least in certain settings, increase inflammatory or brain injury markers like interleukins (IL), NEFL, and S100B [6–10]. High throughput proteomics analyses have similarly suggested that altered peripheral expression levels of S100B or other inflammatory markers can point to early identification of epileptogenesis [11,12]. High levels of inflammatory markers are also associated with a higher risk of epilepsy after stroke [13]. Proinflammatory cytokines, such as IL-1b and IL-6, have been shown to be increased in several studies on epilepsy, with IL-6 associated with seizure severity [2–4,13–16].

We recently reviewed blood biomarkers in epilepsy, clearly a rapidly emerging field [1]. Efforts have so far largely focused on identifying single protein fluid markers. The methodological developments in data analysis, including machine-learning allows additional layers of complexity like search for combination of biomarkers. Neurodegenerative and immune proteins have been identified as seizures reactants, as described above, but are often part of complex biological cascades. A reasonable hypothesis is therefore that a combination of biomarkers has better sensitivity and specificity than single markers. From a bioinformatic perspective, knowledge on protein combination may also allow better pathophysiological understanding.

We recently reported the results of a OLINK proteomic panel in the plasma of individuals (≤50 years) with epilepsy. We identified four proteins that were differentially expressed in persons with recent seizure activity compared to those seizure-free [17], but simple comparisons of expression levels may fail to detect proteins that are useful for classification. In the present study, we re-analysed the dataset with machine learning methods to identify proteins associated with recent seizures. We used two machine learning approaches for feature selection to capture both linear and non-linear associations between the protein expressions and seizure status while using other metadata as explanatory variables including patients' medical diagnostics information.

## Methods

Participants in the study were chosen from the Prospective Regional Epilepsy Database and Biobank for Individualized Clinical Treatment (PREDICT). PREDICT (clinicaltrials.org, NCT04559919) recruits participants from five outpatient facilities in Region Västra Götaland (VGR), Sweden, starting in December 2020. Clinical information is extracted from the medical records at the time of recruitment, and blood samples processed according to the standardized study procedures. The samples were stored at −80°C in Biobank Väst.

For this study, participants who had seizures within the two months before sampling or their last clinic visit and participants with seizure freedom of over one year were selected *(n = 189)*. The Olink Target-96 Neuro-exploratory assay was used to quantify 92 plasma proteins using a dual-recognition immunoassay based on the proximity extension assay (PEA). A comprehensive description of the detailed clinical and experimental approaches leading to the downstream data in this study may be found in our former study [17]. PREDICT started recruitment on 17/11/2020. We included patients meeting our inclusion criteria that had been included up until 23/2/2022. The study is approved by the Swedish Ethical Review Authority (2020–00853) and complied with the principles of the Declaration of Helsinki. All persons provide written informed consent before inclusion in the study.

### Data preprocessing

Data were analyzed in the R programming environment (R version 4.3.2). Proteins with abundances below the level of detection across more than 70% of samples were excluded, leaving 77 proteins for further analysis out of the original count of 92. At the time of experimental analysis, 250 participants had been recruited, of which 243 were initially included in this study. Participants with missing information on seizure status were omitted ($n = 10$). We only included individuals with either seizure freedom for more than one year or with seizures within the two-month period before blood sampling/last clinic visit ($n = 203$). Nine subjects who had a single seizure or multiple seizures without an epilepsy diagnosis were excluded. We identified and omitted five outlier subjects through hierarchical clustering using R built-in functions (R package *stats,* version 4.3.2) (S1 Fig). The downstream analyses were done on the remaining 189 subjects.

### Machine learning and statistical analyses

We used two machine learning approaches, i.e., Sparse Partial Least Squares Discriminant Analysis (sPLS-DA) and Random Forest to capture both linear and non-linear interplays between the protein abundances and the seizure status.

### Random forest

To identify plasma protein markers capable of distinguishing between persons with and without seizures, a Random Forest classifier was employed using R package randomForest (version 4.7–1.1). This ensemble learning method integrates multiple decision trees to improve the predictive accuracy and control overfitting. Each tree was constructed using a bootstrap sample of the data, and at each node, a subset of features was randomly selected to determine the split. The model was built to classify patient status, adjusted for age. The number of trees was set to 1000, at each split in the construction of the trees, 18 variables were randomly selected from the feature set. This was determined based on preliminary grid search-based parameter optimization to improve the model classification performance. Model performance assessment during training and parameter optimization was performed through a 10-fold cross-validation approach using receiver operating characteristic (ROC) as the optimization metric.

### Sparse Partial least squares discriminant analysis (sPLS-DA)

Complementarily, sPLS-DA was used to simultaneously achieve dimension reduction and feature selection, aiming to identify a subset of proteins that most effectively discriminated between the conditions. This method constructs

components that maximize the covariance between the response (disease status) and the predictors (protein levels). The model's performance was assessed using the misclassification Balanced Error Rate (BER) estimates to provide an unbiased prediction error rate.

## Model validation and statistical analysis

Model performance metrics, including accuracy, sensitivity, specificity, and area under the ROC curve, were calculated to assess each model's ability to classify samples accurately. For sPLS-DA, feature importance was calculated using the mean of variable importance (VIP) of the sPLS-DA components. VIP allows ordering the variables, i.e., proteins based on their explanatory power regarding the outcome, i.e., the seizure status. Proteins with VIP > 1 are the most relevant for explaining the response variable being the seizure status [18,19]. Therefore, we chose the top 20 proteins with VIP > 1. Variable importance for the Random Forest model was evaluated using the R package *randomForest* version 4.7–1.2 based on Mean Decrease Accuracy [20]. To keep the balance between the linear and non-linear machine learning-based feature selection, we opted for the same number of the top variables from Random Forest as well, i.e., 20 proteins. The top 20 variable importance for each of the models is illustrated in S2 Fig Statistical analyses were performed using the R statistical software (version 4.0.2). Univariate linear regression was performed on both sets of 20 proteins to evaluate the linear association significance of each protein with seizures as well as their Fold Change between the seizure and seizure-free groups. For evaluating the predictive power of a linear and a non-linear model using individual proteins as well as the combinations of those overlapping with our previous study and those consensus novel candidate protein markers, we used logistic regression (R package *stats,* version 4.3.2) and Support Vector Machine (SVM) (R package *e1071,* version 1.7–16) models [21], respectively, followed by ROC curve evaluations (pROC function from the *pROC* R package, version 1.18.5) for each classifier.

## Results

The demographic and clinical characteristics of the study cohort are summarized in Table 1. To visualize the separation of the two groups we created an sPLS-DA plot (Fig 1) based on the components reflecting the greatest variance. For each seizure and seizure-free group, the center of confidence ellipse was calculated as the mean vector for each group. The covariance matrix was calculated for each group to reflect the shape and orientation of the confidence ellipse. The size of the ellipse for each group was scaled so that it encompasses a 0.95 proportion of the samples based on the chosen 95% confidence level. The complete lists of 77 proteins and their variable importance are in the supplementary materials for reference (S1 and S2 Tables). The proteins identified by machine learning ranked variable importance in both the sPLS-DA and Random Forest models (called consensus proteins) were CDH15, PAEP, LTBP3, PHOSPHO1, NEFL, SFRP1, TDGF1, DUSP3, WWP2 and DSG3. To evaluate the association between the expression levels of each protein and seizures, we conducted univariate linear regression analysis on all 77 proteins (S3 Table). The results of the univariate regression analysis for the top 20 proteins most relevant for classifying the participants as having had a recent seizure or being seizure-free, identified using sPLS-DA and Random Forest based on feature importance, are listed in Table 2. In linear regression, four of the consensus proteins (PAEP, CDH15, LTBP3 and PHOSPHO1) individually had significant associations with the seizure status (p-value < 0.05). Both NEFL and TDGF1 were differentially expressed although not significantly so, but four proteins (SFRP1, DSG3, DUSP3 and WWP2) were not identified in our previous study. For internal validation of the models, the Area Under the Receiver Operating Characteristic Curve (AUC-ROC) for Random Forest (Fig 2A) and sPLS-DA (Fig 2B) were calculated using a 10-fold cross-validation approach. The Random Forest model achieved a higher mean AUC of 0.67 compared to 0.59 for the sPLS-DA model, indicating superior classification performance (Fig 2). Therefore, the top 20 proteins used by Random Forest for classification and their descriptions are listed in Table 3, where the consensus proteins are in bold. For the rest of the analyses, we considered the four consensus proteins found in our previous study (PAEP, CDH15,

**Table 1. Clinical characteristics of the study population.**

| | Seizure-free (>1 year) n=93 | Recent seizures (≤2 months) n=96 |
|---|---|---|
| **Median age (range)** | 44 (19–92) | 44 (18–83) |
| **Gender (%)** | | |
| Male | 40 (43.0%) | 54 (56.3%) |
| Female | 53 (57.0%) | 42 (43.8%) |
| **Median seizure frequency in last 2 months\* (range)** | 0 | 2 (1-60) |
| **Epilepsy type (%)** | | |
| Focal | 41 (44.1%) | 70 (72.9%) |
| Generalized | 26 (28.0%) | 7 (7.3%) |
| Unknown | 26 (28.0%) | 19 (19.8%) |
| **Imaging (%)** | | |
| Normal | 41 (44.1%) | 28 (29.2%) |
| Epileptogenic lesion | 20 (21.5%) | 31 (32.3%) |
| Abnormal, unrelated | 12 (12.9%) | 16 (16.7%) |
| No imaging/unknown | 20 (21.5%) | 21 (21.9%) |
| **Current no. of ASMs\*\* (%)** | | |
| 0 | 3 (3.2%) | 14 (14.6%) |
| 1 | 71 (76.3%) | 30 (31.3%) |
| 2 | 16 (17.2%) | 29 (30.2%) |
| 3 | 3 (3.2%) | 19 (19.8%) |
| 4 | 0 | 3 (3.1%) |
| 5 | 0 | 1 (1.0%) |

\*2 months prior to sampling and/or date of last clinic visit.

\*\* ASM=antiseizure medication.

LTBP3 and PHOSPHO1) and the four consensus proteins that were not identified in our previous study (SFRP1, DSG3, DUSP3 and WWP2).

To compare the discriminatory ability of the combination of the four previously identified significant proteins and the four newly identified consensus proteins with using them individually, we used logistic regression and SVM classifiers. We also included the ROC curve of the model using all eight proteins. Both classifiers, i.e., SVM and logistic regression with all eight proteins outperformed any of the others used individually or in groups of four in classification (Figs 3 and 4). The AUC of the four new consensus proteins (0.76) outperformed the single proteins from the previous study using the SVM classifier (Fig 4). If all the eight proteins were combined, the AUC was 0.9 (Fig 4). Overall, SVM classifiers show higher classification power compared to logistic regression classifiers, highlighting the inherent non-linearity in plasma protein associations with seizures in our cohort (Figs 3 and 4).

## Discussion

We aimed to advance the identification of plasma protein biomarkers associated with seizures in epilepsy by using two complementary machine learning algorithms. We analyzed OLINK proteomics data of patients with epilepsy to identify proteins that contributed to the classification of the patients with recent seizures and seizure freedom. This approach led to the identification of ten consensus candidate protein biomarkers (CDH15, PAEP, LTBP3, PHOSPHO1, NEFL, SFRP1, TDGF1, DUSP3, WWP2 and DSG3), of which four were not identified in our previous study using traditional analyses [17]. Among the identified markers is NEFL, one of the biomarkers most often reported as increased in patients with severe

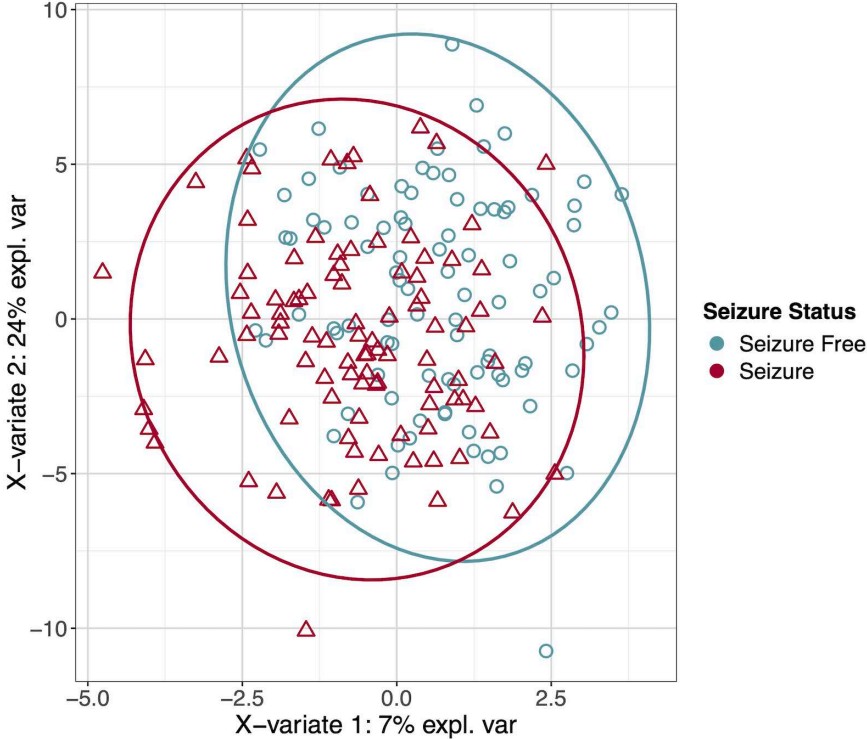

**Fig 1. sPLS-DA plot of the two-dimensional projection of samples based on the components reflecting the greatest variance.** For each group of samples, the mean vector representing the center of the ellipse, and the covariance matrix describing shape and orientation of the ellipse were calculated. The size of the confidence ellipse for each group was scaled according to 95% confidence level to ensure that the ellipse covers the specified proportion of the samples in each group.

seizures [7,27] which validates the method. We then assessed linear and non-linear classification power of the newly identified consensus proteins (SFRP1, DSG3, DUSP3 and WWP2). When all consensus proteins were combined, the AUC was higher for both linear and non-linear classifiers. We also showed that SVM as a non-linear classifier performs better than logistic regression using the same consensus proteins (Figs 3 and 4).

Our results indicate that important biomarkers may be missed or overlooked if single markers are analyzed in isolation, which is an important insight for future proteomic studies in epilepsy. Seizures may be associated with changes in combinations of peripheral markers or expression profiles and our findings indicate the utility of such bioinformatic approaches in biomarker discovery in epilepsy.

There are studies on some of the additional identified consensus proteins in the context of epilepsy or brain development. Secreted Frizzled-Related Protein 1 (SFRP1) modulates the Wnt signaling pathway which is important for neurodevelopment and synaptic plasticity. Dysregulation of Wnt signaling has been suggested to contribute to brain disorders [27,33] with one study indicating decreased expression in the hippocampal tissue of epileptic mice [34]. In a study using a kainic acid-induced rat model, the gene for Calcium-regulated heat-stable protein 1, *Carhsp1*, was identified as a candidate gene associated with epileptogenicity [32].

The other proteins have mainly been reported in inflammation or acute cell injury. DUSP3 (Dual Specificity Phosphatase 3) is a regulator of the MAPK signaling pathway. Recent findings have determined the presence of DUSP3 in the progression of Acute Myocardial Infarction (AMI) and it has been shown that DUSP3 knockdown alleviates oxidative stress, inflammation, and apoptosis [35]. FCAR (Fc Alpha Receptor) binds to the Fc region of immunoglobulins alpha

**Table 2. Univariate analysis of the top 20 proteins identified using the sPLS-DA and Random Forest models based on feature importance. Linear regression analysis was performed on each of the top 20 proteins from both classifiers to evaluate their linear associations with seizures.**

| | sPLS-DA | | | Random forerst | |
|---|---|---|---|---|---|
| **Protein** | **Fold Change** | **P-value** | **Protein** | **Fold Change** | **P-value** |
| **PAEP** | **0.566** | **0.001** | **PAEP** | **0.566** | **0.001** |
| **LTBP3** | **1.251** | **0.014** | **LTBP3** | **1.251** | **0.014** |
| **PHOSPHO1** | **1.101** | **0.024** | **PHOSPHO1** | **1.101** | **0.024** |
| **CDH15** | **1.317** | **0.024** | **CDH15** | **1.317** | **0.024** |
| NEFL | 1.195 | 0.069 | NEFL | 1.195 | 0.069 |
| DUSP3 | 1.261 | 0.09 | DUSP3 | 1.261 | 0.090 |
| IL3RA | 1.09 | 0.106 | TDGF1 | 1.69 | 0.15 |
| ASGR1 | 1.086 | 0.116 | WWP2 | 0.895 | 0.15 |
| FCAR | 1.108 | 0.123 | DSG3 | 0.925 | 0.19 |
| LEPR | 0.927 | 0.131 | AKT1S1 | 0.903 | 0.261 |
| CCL27 | 1.054 | 0.135 | SFRP1 | 0.934 | 0.306 |
| DEFB4A | 1.34 | 0.141 | PTS | 1.11 | 0.31 |
| TDGF1 | 1.69 | 0.15 | EIF4B | 0.948 | 0.326 |
| WWP2 | 0.895 | 0.15 | ILKAP | 0.93 | 0.382 |
| DSG3 | 0.925 | 0.19 | PRTFDC1 | 0.914 | 0.44 |
| TNFRSF13C | 0.923 | 0.247 | ATP6V1F | 0.959 | 0.514 |
| ADAM15 | 0.942 | 0.292 | CETN2 | 0.933 | 0.531 |
| SFRP1 | 0.934 | 0.306 | PLA2G10 | 0.955 | 0.645 |
| GPNMB | 0.979 | 0.399 | RBKS | 1.036 | 0.701 |
| HMOX2 | 0.926 | 0.449 | CARHSP1 | 1.032 | 0.772 |

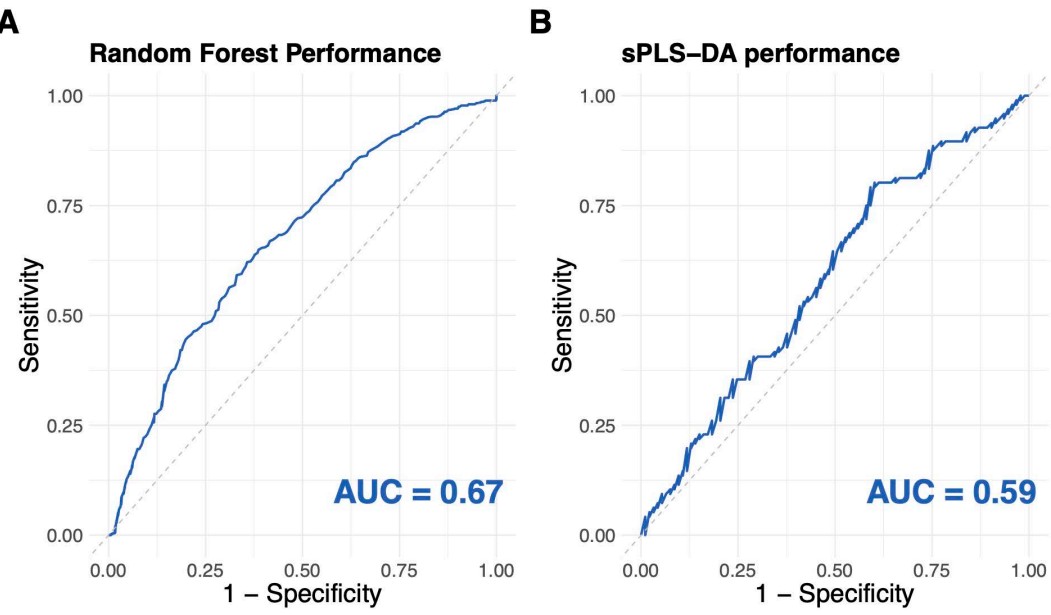

**Fig 2. ROC curves illustrating the classification performance of both classifiers.** For internal validation of the models, AUC for the ROC curves were calculated using a 10-fold cross validation approach. For the Random Forest classifier, the AUC was 0.67 (A) and for the sPLS-DA, the AUC was 0.59 **(B)**.

**Table 3. The top 20 proteins identified using Random Forest are listed. Through a literature review, we identified and cited animal and human studies that have reported associations between these proteins and seizures. The consensus candidate marker proteins are in bold.**

| Uniprot ID | Symbol | Description | Animal | Human |
|---|---|---|---|---|
| **Q9NS15** | **LTBP3** | **Latent-transforming growth factor beta-binding protein 3** | [22] | |
| **P09466** | **PAEP** | **Glycodelin** | | |
| **P55291** | **CDH15** | **Cadherin-15** | | [23,24] |
| **P13385** | **TDGF1** | **Teratocarcinoma-derived growth factor 1** | [25] | |
| **P07196** | **NEFL** | **Neurofilament light polypeptide** | [26] | [7,27,28] |
| **Q8N474** | **SFRP1** | **Secreted frizzled-related protein 1** | [29] | |
| **P32926** | **DSG3** | **Desmoglein-3** | | |
| **Q8TCT1** | **PHOSPHO1** | **Phosphoethanolamine/phosphocholine phosphatase** | [30] | [31] |
| Q9Y2V2 | CARHSP1 | Calcium-regulated heat-stable protein 1 | [32] | |
| Q16864 | ATP6V1F | V-type proton ATPase subunit F | | |
| Q96B36 | AKT1S1 | Proline-rich AKT1 substrate 1 | | |
| Q03393 | PTS | 6-Pyruvoyltetrahydropterin Synthase | | |
| Q9NRG1 | PRTFDC1 | Phosphoribosyltransferase domain-containing protein 1 | | |
| **O00308** | **WWP2** | **NEDD4-like E3 ubiquitin-protein ligase WWP2** | | |
| O15496 | PLA2G10 | Group 10 secretory phospholipase A2 | | |
| Q9H477 | RBKS | Ribokinase | | |
| **P51452** | **DUSP3** | **Dual Specificity Phosphatase 3** | | |
| P41208 | CETN2 | Caltractin (20kD Calcium-Binding Protein) | | |
| Q9H0C8 | ILKAP | ILK Associated Serine/Threonine Phosphatase | | |
| P23588 | EIF4B | Eukaryotic translation initiation factor 4B | | |

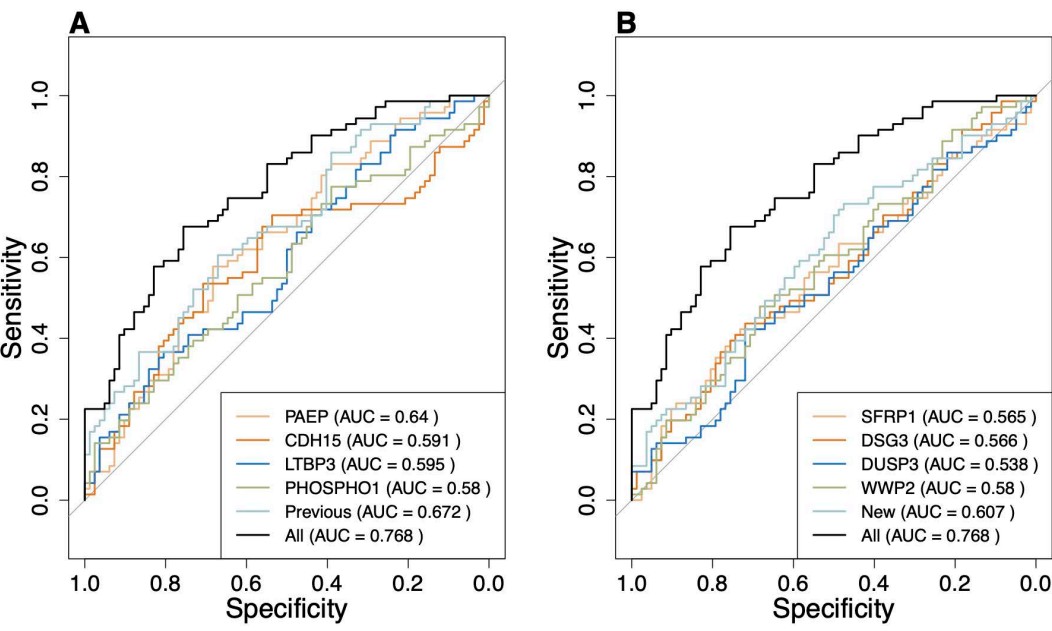

**Fig 3. ROC curves for logistic regression predictive models based on individual and combined four protein levels that were significantly differentially expressed between seizure and seizure-free individuals (A) and the new consensus proteins identified by the machine learning workflow (B).** The AUC for the combination of all eight consensus proteins (0.76) is greater than the individual AUCs of the proteins as predictors for the seizure status. The ROC curve for the eight consensus proteins is indicated as "All" in the legend and is identically added to both A and B for better comparison with the rest of the curves.

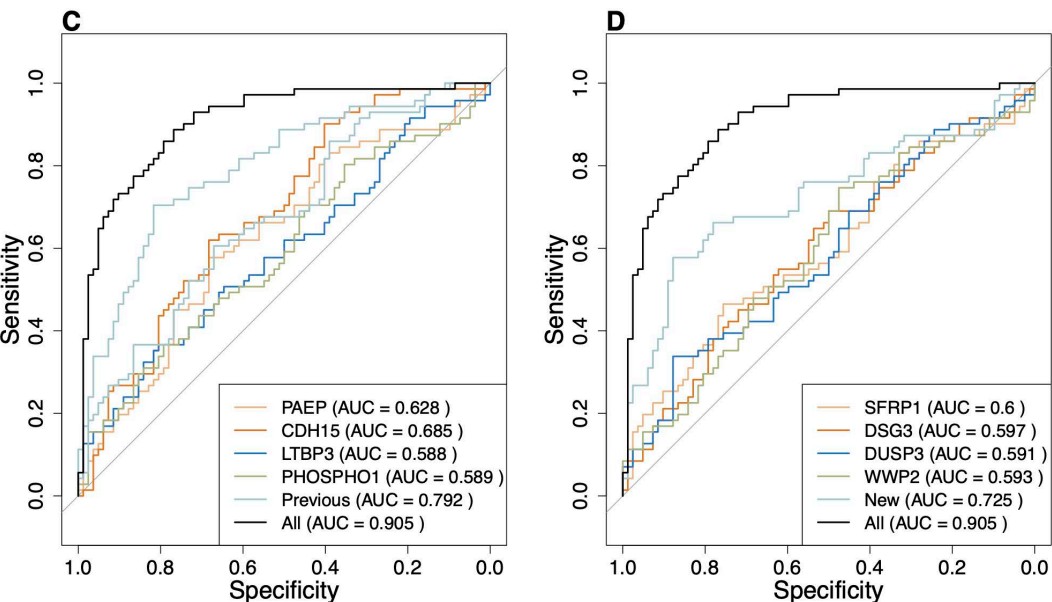

**Fig 4. ROC curves for SVM predictive models based on individual and combined levels of four proteins that were significantly differentially expressed between seizure and seizure-free individuals (A) and the additional consensus proteins identified by the machine learning workflow (B).** The AUC for the combination of all eight consensus proteins (0.90) is greater than the individual AUCs of the proteins as predictors for the seizure status. The ROC curve for the eight consensus proteins is indicated as "All" in the legend and is identically added to both A and B for better comparison with the rest of the curves. Using the same training data SVM demonstrates higher classification power compared with the logistic regression predictive models, corroborating the inherent non-linearity in protein associations in this study.

and mediates several functions including cytokine, superoxide radicals and inflammatory radicals production [36]. DSG3 (Desmoglein 3) is a cell-cell adhesion molecule primarily studied in the context of skin diseases [37]. Finally, WWP2 (WW Domain Containing E3 Ubiquitin Protein Ligase 2) is involved in the ubiquitin-proteasome system. WWP2 knockout mice have been shown to have an increased innate immune response [38]. Taken together, this is neurobiologically intriguing as inflammation is repeatedly identified as both a cause of epileptogenesis and a reflection of disease activity [39].

Methodologically, the combined use of Random Forest and sPLS-DA in our analysis was an attempt to capture both linear and non-linear relationships in the data and determine if protein expression levels could hold classifying potential beyond comparing levels of individual proteins. Proteomic profiling in routine clinical protocols is not yet possible, but many inflammatory mediators like interleukins are routinely measured in rheumatology and immunology. It may soon be possible to improve epilepsy care by enabling more precise and personalized treatment approaches. Some patients with epilepsy refractory to traditional epilepsy medications respond to immunotherapy, suggesting that a better understanding of immune pathways on an individual level is likely to be integral to epilepsy care in the future [39].

To address the complex relationships and interactions between seizure status and protein NPX levels, we opted for Random Forest over gradient boosting algorithms, such as XGBoost, due to its resilience against noise, reduced risk of overfitting, and enhanced interpretability of feature importance. The bagging-based methodology of Random Forest ensures stable and biologically meaningful feature selection, particularly in high-dimensional proteomic datasets [40]. Nevertheless, Random Forest exhibits inherent challenges regarding the reproducibility of feature importance rankings. The algorithm's reliance on randomized processes, including bootstrapped samples and random feature selection at each tree node, can lead to variability in feature importance scores across runs. This inconsistency is particularly pronounced in datasets with high dimensionality and intercorrelated features, such as proteomics data.

To address this issue, we incorporated a fixed random seed before each Random Forest run, ensuring consistent and reproducible outputs. Furthermore, all computational analyses were conducted under uniform machine specifications to eliminate variability stemming from differences in computational environments. These measures enhanced the reproducibility and reliability of our findings, bolstering confidence in the robustness of the identified biomarkers. In addition, sPLS-DA was employed as a complementary approach to Random Forest, capitalizing on its ability to perform simultaneous feature selection and dimensionality reduction. By emphasizing linear associations, sPLS-DA generates interpretable models that isolate a concise set of proteins related to seizure status while reducing overfitting through sparsity. However, sPLS-DA has limitations, including sensitivity to parameter tuning, reliance on linearity assumptions, and potential exclusion of features with moderate effects. These constraints highlight the importance of integrating sPLS-DA with nonlinear methods, such as Random Forest, to provide a comprehensive analysis of proteomic signatures associated with seizures.

Clinically, there is a great need for blood biomarkers in epilepsy, a field that trails other nervous system disorders like dementia and traumatic brain injury. Currently, seizures can only be detected by patient or caregiver interview and seizure diaries, which are crude and unreliable methods. Seizures are also very crude markers of underlying brain disease.[41] Other potential biomarkers like EEG or CSF analyses are laborious, invasive, or difficult to scale – so blood tests useful for the study and monitoring of epilepsy are greatly needed. Hopefully, machine-learning bioinformatic methods will allow identification of proteins or patterns of proteins that could serve as the basis for developing blood-based tests to monitor disease progression, predict seizure risk, and personalize treatment strategies. The involvement of the herein identified proteins in diverse biological processes, from synaptic plasticity and neuroinflammation to immune responses, reflects the complex and multifactorial nature of epilepsy, emphasizing the need for a multidisciplinary approach.

## Conclusions

We identified additional candidate plasma protein biomarkers associated with seizures in persons with epilepsy using consensus machine learning techniques. In addition to the four previously reported proteins with significantly different levels, the consensus machine learning approach prioritized using six additional proteins for classification. Using eight candidate markers together corroborates the utility of proteomic profiling in epilepsy research and highlights the importance of studying profiles of marker expression in addition to individual proteins. Further studies are needed to validate these findings in larger, independent cohorts and to explore the mechanistic roles of these candidate proteins in epilepsy pathogenesis.

## Supporting information

**S1 Fig. Hierarchical clustering dendrogram to identify outlier subjects.** The distance measure was "Euclidean distance" for calculating the distance matrix. The agglomeration method for the hierarchical clustering function was "Average". The clustering dendrogram shows that the outliers are separate from the main cluster at a height of greater than 10. (TIF)

**S2 Fig. Variable importance visualization for both models including the top 20 variables.** After training the sPLS-DA model, we chose the top explanatory variables (proteins) with variable importance (VIP) > 1. Then to keep the balance between linearity and non-linearity of associations with the seizure status, we chose the same number of top VIP-ranked explanatory variables (proteins) from the trained Random Forest model (20 proteins). Finally, we chose the intersection of the mentioned top explanatory variables (and called them consensus proteins) which are shown to be important (using the ranked VIP) in discriminating between seizure and seizure-free states. (TIF)

**S1 Table. sPLS-DA VIPs.** For each protein the variable importance was calculated using the mean of VIPs across the sPLS-DA components. (CSV)

**S2 Table. Random Forest VIPs.** For the Random Forest model VIPs were evaluated for each protein based on Mean Decrease Accuracy.
(XLSX)

**S3 Table. Univariate analysis of all 77 proteins.** Linear regression analysis was performed on each of the 77 proteins to evaluate the linear associations between them and seizures.
(XLSX)

## Acknowledgments

This study was conducted using professional biobank services from Biobank West and Biobank Sweden.

## Author contributions

**Conceptualization:** Rakesh Kumar Banote, Johan Zelano.

**Data curation:** Saman Hosseini Ashtiani, Sarah Akel, Fredrik Asztely.

**Formal analysis:** Saman Hosseini Ashtiani, Sarah Akel, Fredrik Asztely, Johan Zelano.

**Funding acquisition:** Johan Zelano.

**Investigation:** Saman Hosseini Ashtiani, Sarah Akel, Johan Zelano.

**Methodology:** Saman Hosseini Ashtiani, Rakesh Kumar Banote.

**Project administration:** Johan Zelano.

**Resources:** Johan Zelano.

**Software:** Saman Hosseini Ashtiani.

**Supervision:** Saman Hosseini Ashtiani, Johan Zelano.

**Validation:** Saman Hosseini Ashtiani, Sarah Akel, Johan Zelano.

**Visualization:** Saman Hosseini Ashtiani, Sarah Akel.

**Writing – original draft:** Saman Hosseini Ashtiani, Sarah Akel, Johan Zelano.

**Writing – review & editing:** Saman Hosseini Ashtiani, Johan Zelano.

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
