## [Decision Letter · Decision Letter 0]

Dear Dr. Hosseini Ashtiani,

After careful consideration by 2 Reviewers and an Academic Editor, all of the critiques of the Reviewers must be addressed in detail in a revision to determine publication status. If you are prepared to undertake the work required, I would be pleased to reconsider my decision, but revision of the original submission without directly addressing the critiques of the Reviewers does not guarantee acceptance for publication in PLOS ONE. If the authors do not feel that the queries can be addressed, please consider submitting to another publication medium. A revised submission will be sent out for re-review. The authors are urged to have the manuscript given a hard copyedit for syntax and grammar.

We look forward to receiving your revised manuscript.

Kind regards,

Stephen D. Ginsberg, Ph.D.

Section Editor

PLOS ONE

“This study was funded by grants from Jeansson foundation, Swedish Society for Medical Research (SS18-0040), Region Västra Götaland (VGFOUREG-968476), Swedish state under the agreement between the Swedish government and the county councils and the ALF-agreement (715781).”

“J.Z. received speaker honoraria from UCB and Eisai for non-branded education events; and as employee of Sahlgrenska University Hospital is or has been an investigator/sub investigator in clinical trials sponsored by GW Pharma, SK life science, UCB, Angilini Pharma, and Bial (no personal compensation).”

We note that you received funding from a commercial source: UCB and Eisai

6. We note that you have indicated that there are restrictions to data sharing for this study. PLOS only allows data to be available upon request if there are legal or ethical restrictions on sharing data publicly. For more information on unacceptable data access restrictions, please see http://journals.plos.org/plosone/s/data-availability#loc-unacceptable-data-access-restrictions.

**Comments to the Author**

1. Is the manuscript technically sound, and do the data support the conclusions?

Reviewer #1: Yes

Reviewer #2: Partly

2. Has the statistical analysis been performed appropriately and rigorously?

Reviewer #1: Yes

Reviewer #2: Yes

3. Have the authors made all data underlying the findings in their manuscript fully available?

Reviewer #1: Yes

Reviewer #2: Yes

4. Is the manuscript presented in an intelligible fashion and written in standard English?

Reviewer #1: Yes

Reviewer #2: No

Reviewer #1: The article is original and very relevant for the field. This is a comprehensive metabolomics approach which analyzed OLINK proteomics data from a large epilepsy cohort in which they have previously found four differentially expressed proteins (CDH15, PAEP, LTBP3, PHOSPHO1) in patients with recent seizures. Using two machine-learning techniques, they confirmed the four previously detected proteins, but additionally uncovered ten consensus candidate protein biomarkers (CDH15, PAEP, LTBP3, PHOSPHO1, NEFL, SFRP1, TDGF1, DUSP3, WWP2 and DSG3) that contributed to classification of patients as being seizure-free or not.

The methodology of the study is modern and very complex, using very modern tools for biochemistry analysis and molecular biology.

The results conducted to the new findings showing that: using a combination of the four differentially expressed proteins (CDH15, PAEP, LTBP3, PHOSPHO1) as well as the additional consensus proteins (SFRP1, DSG3, DUSP3 and WWP2) not present among their previous findings out performed individual proteins in identifying individuals with recent seizures, highlighting the potential of multi-protein profiles.

The conclusions are consistent with the evidence and arguments presented.

The references are appropriate, including some very relevant authors experience in the field. I recommend some corrections.

1. Number of lines would be very usefull for small pointed corrections.

2. Both Introduction and Discussions should be more detailed to include the other biomarkers for epilepsy.

For Introduction you may see also M. Musteata et al., 2013, The 1H NMR Profile of Healthy Dog Cerebrospinal Fluid, PLoS ONE 8(12): e81192. doi:10.1371/journal.pone.0081192 M. Armaşu, R.M.A. Packer, S. Cook, G. Solcan, H.A. Volk, An exploratory study using a statistical approach as a platform for clinical reasoning in canine epilepsy, The Veterinary journal, 2014, 202, 292-296, DOI: 10.1016/j.tvjl.2014.08.008

Reviewer #2: This study used machine learning (Random Forest and sPLS-DA) to analyze plasma proteomics data from individuals with epilepsy, aiming to identify biomarkers linked to recent seizures. Ten candidate proteins were identified, including four previously known and six novel markers. The authors suggest that combination of proteins improved seizure classification accuracy, highlighting the potential for blood-based biomarkers in epilepsy monitoring. The study is relevant as there is an unmet need for biomarkers however there are certain aspects of the work that need improvement to increase the strength of the manuscript and comprehension of the work for the reader.

1. Review manuscript for grammatical or typographical errors. Key sentences in the manuscript need to be reviewed to be written in a clearer and more concise way.

-The title of the manuscript has typos.

-This sentence has grammar errors -” Using the four candidate markers together corroborate the utility of proteomic profiling in epilepsy research and highlight the importance of studying profiles of marker expression in addition to individual proteins. “

- Page 9, last sentence has a typo.

- This sentence has grammar errors - “Some patients with epilepsy refractory to traditional epilepsy medications respond to immunotherapy, suggesting that a better understanding of immune pathways on an individual level are likely to be integral to epilepsy care in the future.”

-Key sentence to be written clearly –“ Four proteins (PAEP, CDH15, LTBP3 and PHOSPHO1) were demonstrated to have significant association with the seizure status (p-value < 0.05) using both sPLS-DA and Random Forest models besides the rest of the proteins used for classification in both classifiers that were not among our findings in the previous study (SFRP1, DSG3, DUSP3 and WWP2).”

2. Stronger reasoning is necessary for the selection of the 4 novel proteins (SFRP1, DUSP3, WWP2 and DSG3 ) since they were not significantly differentially expressed in any of the methods - sPLS-DA and Random Forest.

3. In the discussion the authors bring up “FCAR” related to inflammation, but it has not been mentioned or discussed in the data before.

- “FCAR (Fc Alpha Receptor) Binds to the Fc region of immunoglobulins alpha and mediates several functions including cytokine, superoxide radicals and inflammatory radicals.” production.

4. More information is needed in the figure descriptions and tables to include statistical tests performed for the analysis. Additionally, the AUC in the description of Figure 2 do not match with the AUC in the figure. Label correctly the figures when attaching.

5. In the discussion of Figure 3, the authors mention that the combination of four proteins outperformed any of these previously identified proteins when used individually to predict individuals with recent seizures. However, in Figure 3B, DUSP3 has an AUC=1 suggesting that the protein perfectly distinguishes between seizure and seizure-free individuals without any error. This is either a typo or an issue in the analysis.

**Do you want your identity to be public for this peer review?**  For information about this choice, including consent withdrawal, please see our Privacy Policy

Reviewer #1: **Yes: ** G. Solcan

Reviewer #2: No

---

## [Author Response · Author response to Decision Letter 1]

22 May 2025

Manuscript Number: PONE-D-24-55733

Dear Dr. Ginsberg,

Thank you for the decision letter and the reviewer comments. We have made multiple clarifications and added new analyses as suggested, which has in our opinion substantially improved the paper. Specifically, we now explain the rationale for the selected markers and have added a new AUC analysis showing how much discriminative value they add in comparison to the markers previously identified in traditional analyses. The new analysis originates in a most useful reviewer suggestion; and illustrates both how the machine learning-approach found additional proteins beyond the standard analyses and that single marker analysis may not suffice in future proteomics studies in epilepsy. We have addressed all reviewer comments and the comments from the editorial office. We have also altered the abstract to reflect the new results.

For the authors,

Saman Hosseini Ashtiani, Ph.D.

Specifically, we have made the following changes:

Reviewer #1:

The article is original and very relevant for the field. This is a comprehensive metabolomics approach which analyzed OLINK proteomics data from a large epilepsy cohort in which they have previously found four differentially expressed proteins (CDH15, PAEP, LTBP3, PHOSPHO1) in patients with recent seizures. Using two machine-learning techniques, they confirmed the four previously detected proteins, but additionally uncovered ten consensus candidate protein biomarkers (CDH15, PAEP, LTBP3, PHOSPHO1, NEFL, SFRP1, TDGF1, DUSP3, WWP2 and DSG3) that contributed to classification of patients as being seizure-free or not. The methodology of the study is modern and very complex, using very modern tools for biochemistry analysis and molecular biology. The results conducted to the new findings showing that: using a combination of the four differentially expressed proteins (CDH15, PAEP, LTBP3, PHOSPHO1) as well as the additional consensus proteins (SFRP1, DSG3, DUSP3 and WWP2) not present among their previous findings out performed individual proteins in identifying individuals with recent seizures, highlighting the potential of multi-protein profiles.

The conclusions are consistent with the evidence and arguments presented.

The references are appropriate, including some very relevant authors experience in the field. I recommend some corrections.

Comment 1:- Number of lines would be very useful for small pointed corrections.

Response: Thank you for the encouraging evaluation. Number of lines have been added throughout the manuscript.

Comment 2: Both Introduction and Discussions should be more detailed to include the other biomarkers for epilepsy. For Introduction you may see also M. Musteata et al., 2013, The 1H NMR Profile of Healthy Dog Cerebrospinal Fluid, PLoS ONE 8(12): e81192. doi:10.1371/journal.pone.0081192 M. Armaşu, R.M.A. Packer, S. Cook, G. Solcan, H.A. Volk, An exploratory study using a statistical approach as a platform for clinical reasoning in canine epilepsy, The Veterinary journal, 2014, 202, 292-296, DOI: 10.1016/j.tvjl.2014.08.008

Response:

We have added more detail to the discussion:

- Page 10, 512-514 and 526-528

- Page 11, 573-575

We added the latter of the suggested references to the discussion (page 11, 575).

Reviewer #2:

This study used machine learning (Random Forest and sPLS-DA) to analyze plasma proteomics data from individuals with epilepsy, aiming to identify biomarkers linked to recent seizures. Ten candidate proteins were identified, including four previously known and six novel markers. The authors suggest that combination of proteins improved seizure classification accuracy, highlighting the potential for blood-based biomarkers in epilepsy monitoring. The study is relevant as there is an unmet need for biomarkers however there are certain aspects of the work that need improvement to increase the strength of the manuscript and comprehension of the work for the reader.

Comments:

Comment 1:* Review manuscript for grammatical or typographical errors. Key sentences in the manuscript need to be reviewed to be written in a clearer and more concise way.

-The title of the manuscript has typos.

-This sentence has grammar errors -” Using the four candidate markers together corroborate the utility of proteomic profiling in epilepsy research and highlight the importance of studying profiles of marker expression in addition to individual proteins. “

- Page 9, last sentence has a typo.

- This sentence has grammar errors - “Some patients with epilepsy refractory to traditional epilepsy medications respond to immunotherapy, suggesting that a better understanding of immune pathways on an individual level are likely to be integral to epilepsy care in the future.”

-Key sentence to be written clearly –“ Four proteins (PAEP, CDH15, LTBP3 and PHOSPHO1) were demonstrated to have significant association with the seizure status (p-value < 0.05) using both sPLS-DA and Random Forest models besides the rest of the proteins used for classification in both classifiers that were not among our findings in the previous study (SFRP1, DSG3, DUSP3 and WWP2).”

Response:

The grammatical and typo errors have been fixed including:

- The title’s typo has been fixed

- Page 11, 572-574

- Page 9, 413-414

- Page 10, 537

- Pages 5 and 6, 247-292

Comment 2:* Stronger reasoning is necessary for the selection of the 4 novel proteins (SFRP1, DUSP3, WWP2 and DSG3 ) since they were not significantly differentially expressed in any of the methods - sPLS-DA and Random Forest.

Response:

We have now clarified this very important point:

- Pages 5, 215-228

- Pages 5 and 6, 247-292

- Pages 8 and 9, 366-384

Comment 3:* In the discussion the authors bring up “FCAR” related to inflammation, but it has not been mentioned or discussed in the data before. “FCAR (Fc Alpha Receptor) Binds to the Fc region of immunoglobulins alpha and mediates several functions including cytokine, superoxide radicals and inflammatory radicals.” production.

Response:

This was a residue from a previous manuscript version, and has been removed.

Comment 4:* More information is needed in the figure descriptions and tables to include statistical tests performed for the analysis. Additionally, the AUC in the description of Figure 2 do not match with the AUC in the figure. Label correctly the figures when attaching.

Response:

The figure captions and table descriptions have been corrected and/or improved:

- Figure 1 caption: Page 6, 294-299

- Figure 2 caption: Page 6, 301-304

- Table 2 caption: Page 7, 342-345

- Table 3 caption: Page 8, 353-355

- Figure 3 caption: Page 9, 386-392

- Figure 4 caption: Page 9, 395-404

- Supporting information captions: 754-778

Comment 5:* In the discussion of Figure 3, the authors mention that the combination of four proteins outperformed any of these previously identified proteins when used individually to predict individuals with recent seizures. However, in Figure 3B, DUSP3 has an AUC=1 suggesting that the protein perfectly distinguishes between seizure and seizure-free individuals without any error. This is either a typo or an issue in the analysis.

Response:

This was a plotting error, which has been fixed (Figure 3)

6. PLOS authors have the option to publish the peer review history of their article (what does this mean?). If published, this will include your full peer review and any attached files.

Do you want your identity to be public for this peer review? For information about this choice, including consent withdrawal, please see our Privacy Policy.

Reviewer #1: Yes: G. Solcan

Reviewer #2: No

Response:

Yes, we have no objection to the reviews being published. It shows very nicely how the additional analysis of all biomarkers added to the manuscript.

We have also addressed the following editorial comments.

Response:

We have updated the format and styles of the manuscript.

Response:

All the codes will be publicly available via GitHub repository after the probable acceptance.

Response:

We have made sure that the grant numbers are correct and consistent.

“This study was funded by grants from Jeansson foundation, Swedish Society for Medical Research (SS18-0040), Region Västra Götaland (VGFOUREG-968476), Swedish state under the agreement between the Swedish government and the county councils and the ALF-agreement (715781).”

Response:

The amended statement has been added within the cover letter.

“J.Z. received speaker honoraria from UCB and Eisai for non-branded education events; and as employee of Sahlgrenska University Hospital is or has been an investigator/sub investigator in clinical trials sponsored by GW Pharma, SK life science, UCB, Angilini Pharma, and Bial (no personal compensation).”

We note that you received funding from a commercial source: UCB and Eisai

Response:

The amended statement has been added within the cover letter.

6. We note that you have indicated that there are restrictions to data sharing for this study. PLOS only allows data to be available upon request if there are legal or ethical restrictions on sharing data publicly. For more information on unacceptable data access restrictions, please see http://journals.plos.org/plosone/s/data-availability#loc-unacceptable-data-access-restrictions.

Response:

Data Availability Statement has been added to page 15: 783-791.

The underlying data in this study are sensitive personal data according to the Swedish Data Protection Regulation. The ethical approval of the Swedish Ethical Review Authority for this study (2020-853) states that the data and results can only be published on a group level, i.e., without any patient-level details. Both original and translated ethical approvals are uploaded. The contact details for the Swedish Ethical Review Authority that has imposed such restrictions along with a link to clear explanations identifying any patient-level health-related data as sensitive:

registrator@etikprovning.se

Phone: +46104750800

https://www.imy.se/en/organisations/data-protection/data-protection-within-different-areas/processing-of-personal-data--for-researchers/

Post address:

Etikprövningsmyndigheten

Box 2110

750 02 Uppsala

Response:

The information has been added to lines 157-159 of the “Methods” section.

Response:

The information has been added to lines 157-159 of the “Methods” section.

The full response and amended statements have been included within the response and cover letter files.

Sincerely,

For the authors,

Saman Hosseini Ashtiani, Ph.D.

---

## [Decision Letter · Decision Letter 1]

Identification of plasma proteins associated with seizures in epilepsy: A consensus machine learning approach

PONE-D-24-55733R1

Dear Dr. Hosseini Ashtiani,

We’re pleased to inform you that your manuscript has been judged scientifically suitable for publication and will be formally accepted for publication once it meets all outstanding technical requirements.

Kind regards,

Stephen D. Ginsberg, Ph.D.

Section Editor

PLOS ONE

**Comments to the Author**

Reviewer #1: All comments have been addressed

Reviewer #2: All comments have been addressed

2. Is the manuscript technically sound, and do the data support the conclusions?

Reviewer #1: Yes

Reviewer #2: Yes

3. Has the statistical analysis been performed appropriately and rigorously?

Reviewer #1: Yes

Reviewer #2: Yes

4. Have the authors made all data underlying the findings in their manuscript fully available?

Reviewer #1: Yes

Reviewer #2: Yes

5. Is the manuscript presented in an intelligible fashion and written in standard English?

Reviewer #1: Yes

Reviewer #2: Yes

Reviewer #1: The authors have made substantial Improvements of the manuscript, including all the corrections suggested.

Reviewer #2: Thank you to the authors for reviewing the manuscript and addressing the comments that were made. The manuscript looks better now.

**Do you want your identity to be public for this peer review?** For information about this choice, including consent withdrawal, please see our Privacy Policy

Reviewer #1: No

Reviewer #2: No

---

## [Editor Report · Acceptance letter]

PONE-D-24-55733R1

PLOS ONE

Dear Dr. Hosseini Ashtiani,

I'm pleased to inform you that your manuscript has been deemed suitable for publication in PLOS ONE. Congratulations! Your manuscript is now being handed over to our production team.

Kind regards,

on behalf of

Dr. Stephen D. Ginsberg

Section Editor

PLOS ONE